# Excitatory Neurons Derived from Human-Induced Pluripotent Stem Cells Show Transcriptomic Differences in Alzheimer’s Patients from Controls

**DOI:** 10.3390/cells12151990

**Published:** 2023-08-02

**Authors:** Ram Sagar, Ioannis Azoidis, Cristina Zivko, Ariadni Xydia, Esther S. Oh, Paul B. Rosenberg, Constantine G. Lyketsos, Vasiliki Mahairaki, Dimitrios Avramopoulos

**Affiliations:** 1Department of Genetic Medicine, Johns Hopkins University School of Medicine, Baltimore, MD 21205, USA; 2The Richman Family Precision Medicine Center of Excellence in Alzheimer’s Disease, Johns Hopkins University School of Medicine, Baltimore, MD 21287, USA; 3Department of Medicine, Johns Hopkins University School of Medicine, Baltimore, MD 21224, USA; 4Department of Psychiatry and Behavioral Sciences, Johns Hopkins University School of Medicine, Baltimore, MD 21287, USA

**Keywords:** iPSCs, Alzheimer’s disease, excitatory neurons, transcriptomics

## Abstract

The recent advances in creating pluripotent stem cells from somatic cells and differentiating them into a variety of cell types is allowing us to study them without the caveats associated with disease-related changes. We generated induced Pluripotent Stem Cells (iPSCs) from eight Alzheimer’s disease (AD) patients and six controls and used lentiviral delivery to differentiate them into excitatory glutamatergic neurons. We then performed RNA sequencing on these neurons and compared the Alzheimer’s and control transcriptomes. We found that 621 genes show differences in expression levels at adjusted *p* < 0.05 between the case and control derived neurons. These genes show significant overlap and directional concordance with genes reported from a single-cell transcriptome study of AD patients; they include five genes implicated in AD from genome-wide association studies and they appear to be part of a larger functional network as indicated by an excess of interactions between them observed in the protein–protein interaction database STRING. Exploratory analysis with Uniform Manifold Approximation and Projection (UMAP) suggests distinct clusters of patients, based on gene expression, who may be clinically different. Our research outcomes will enable the precise identification of distinct biological subtypes among individuals with Alzheimer’s disease, facilitating the implementation of tailored precision medicine strategies.

## 1. Introduction

Alzheimer’s disease (AD) and dementia globally affect more than 55 million people, and the cases are projected to triple by 2050 [1], causing a major and worsening public health crisis.

Even with the newly approved drugs (lecanemab and donanemab), there is no clinically effective treatment for AD. Lecanemab only minimally delays progression of AD symptoms, and does not fully treat the disease. The newly approved drug lecanemab only minimally delays progression of AD symptoms, and does not fully treat the disease. Lacking disease therapies that reverse or substantially delay progression of AD symptoms, it is important to continue to improve our understanding of its biology including both genetic and environmental contributions. Despite significant progress in the AD genetics, including the identification of three genes (*PSEN1*, *PSEN2*, and *APP*) [2,3,4] that cause familial AD, the discovery of *APOE* as a major risk factor [5,6] and the identification of over 30 other genes with smaller but significant contributions [7,8,9,10], the etiological treatments currently approved have not shown substantial improvements in managing the disease progress. A promising solution to this problem may be the application of precision medicine to allow for a better match between patients and candidate treatments. At the Johns Hopkins Richman Family Precision Medicine Center of Excellence in Alzheimer’s Disease (JH-AD-PMC), we are continuously looking for new ways to achieve this goal.

One of the most promising ways to accurately characterize patients in order to identify biomarkers that can help towards the goal of precision medicine is the study of their biological materials. Among the samples and materials most likely to provide answers are the cells directly involved in the disease under study. For diseases of the human brain such as AD, this poses a significant problem, as there are multiple limitations in the study of brain cells. The brain is not easily accessible and cannot be sampled without risk. Even if sampled, a mix of cells would be acquired (neurons, astrocytes, microglia etc.) at varying relative abundance. Further, these cells would have been subjected not only to the ongoing disease process, but also to the medications used against it and to many unknown environmental variables. Recent advances in cell engineering have opened new possibilities for brain tissue at the individual level. Accessible cells such as skin fibroblasts or peripheral blood mononuclear cells (PMNCs) can be easily acquired and then reprogrammed into iPSCs, erasing the epigenetic marks of differentiation, and allowing new differentiation to a variety of cell types [11,12,13,14]. Induced PSCs were first established more than a decade ago using the four Yamanaka transcription factors (OCT3/4, SOX2, KLF4, c-MYC) to reprogram mouse adult fibroblast cells [15]. Since then, different technological advances have contributed to reliably establish human iPSCs from different cells, including new combinations of transcription factors, using small molecules, or reprogramming with episomal vectors [16]. Advances in differentiation methods have allowed the generation of multiple cell types from iPSCs, including many types of neuronal and glial cells [17,18] that resemble in vivo counterparts; these have become a popular tool for preclinical disease modeling in scientific research [17,19]. At the same time, advances in DNA sequencing allow for the accurate characterization of the transcriptomic state of cells and tissues, a window into their metabolic state and genomic make-up.

Here we investigate whether cells derived from iPSC of individual AD patients show transcriptomic differences when compared to cells derived from cognitively unimpaired individuals, after they have been first reprogrammed to pluripotency followed by differentiation into excitatory glutamatergic neurons. Such differences could not only be useful in predicting disease, but also in categorizing patients into different evidence-based clusters that might predict their course and response to treatment, a big step forward for precision medicine in AD.

## 2. Materials and Methods

### 2.1. Patients and Controls

Blood from patients (JHU-AD-01 to -06 and JHU-AD-08), as well as from healthy individuals (JHU-WT-03, -04, -07, and -08) was collected through the Johns Hopkins Alzheimer’s Disease Research Center (ADRC) and the Johns Hopkins Memory and Alzheimer’s Treatment Center (MATC). Blood from patient JHU-AD-07 was provided to us from collaborators through the ongoing S-CitAD clinical trial (ClinicalTrials.gov Identifier: NCT03108846). Patients recruited in the latter study had AD dementia, Mini-Mental State Examination (MMSE) scores of 5–26, and met criteria for agitation syndrome [20]. Adult skin fibroblasts from cognitively unimpaired individuals (JHU-WT-01 and -05) were obtained from the Coriell Institute for Medical Research [21]. AD patients had a mean age of 71.4 years and controls had a mean age of 71.7 years. Table 1 shows the age and sex of all participants. The participants were overwhelmingly female with only 3 males among the patients. Blood was collected by venipuncture into yellow-top tubes (containing trisodium citrate, citric acid, and dextrose) and shipped to the Johns Hopkins Genetics Core facility for the isolation of PMNCs which were then stored in liquid nitrogen until used. All patients had sporadic AD.

This study was approved by the Johns Hopkins Institutional Review Board (IRB).

### 2.2. Induced Pluripotent Stem Cell (iPSC) Generation, Culture and Maintenance

Peripheral blood mononuclear cells (PMNCs) were isolated from the blood samples of the individuals who consented to their participation in the study. They were then reprogrammed to iPSCs by using a transient expression method (nucleofection) involving three plasmid vectors (MOS, MMK and GBX) under feeder-free/xeno-free culture on 4D Nucleofector (Lonza, Basel, Switzerland) [12,22,23,24]. Generated iPSCs were characterized by immunocytochemistry for pluripotency markers, including Nanog, OCT4 and TRA-1-60, flow cytometry, and karyotyping. Established cell lines were cultured (250 K cells/well in 6 well plate) on vitronectin-coated tissue culture plates in Essential-8 (E8) medium with 10 uM Rock inhibitor (Y-27632) during seeding and then maintained in E8 medium till 80–90% confluency [12,23].

### 2.3. Lentivirus Transduction of iPSCs with NgN2

To acquire a high yield of functional neurons, we transduced the generated human iPSCs with Ngn2 and rTTA expressing lentivirus (lentivirus was purchased by Cellomics Technology, Arbutus, MD, USA). We followed an established protocol for the generation of induced neuronal cells in 21 days [18,25]. Once we established iPSCs confluency at 40–50%, the cells were transduced with Ngn2 (1.5 uL) and rtTA (1.5 uL) expressing lentivirus. Further polybrene (1 ug/mL, Santa Cruz, CA, USA) was added at the time of transduction and cells were incubated for 6 h at 37 °C. After 6 h, culture medium was replaced with fresh E8 medium and cells were incubated for 24 h or until 90% confluency was achieved. Cells were hereafter passaged only in rock inhibitor supplemented E8 medium. Transduced cells (250 k cells/well in a 6-well plate) were used for neuronal differentiation in this study.

### 2.4. Neuronal Differentiation of Ngn2 Transduced iPSCs

The iPSCs were differentiated into glutamatergic excitatory neurons as described in a previously published protocol [18,25,26]. Ngn2-transduced cells (250 K cells/well) were plated on vitronectin coated 6-well culture plates. After 48 h, culture medium was replaced with fresh E8 medium. To start the differentiation, Doxycycline was added to induce Ngn2 expression (Day 0) using the iN-N2 induction medium {DMEM/F12 (Gibco, Grand Island, NY, USA), N-2 supplement (Gibco, Grand Island, NY, USA), D-Glucose (Millipore Sigma, St. Louis, MO, USA), 2-β-mercaptoethanol (Gibco, Grand Island, NY, USA), Primocin (Invivogen, San Diego, CA, USA), BDNF (10 ng/mL, Peprotech, Cranbury, NJ, USA), NT3 (10 ng/mL, Peprotech, Cranbury, NJ, USA), Laminin (200 ng/mL, Millipore Sigma, St. Louis, MO, USA), and Doxycycline (2 µg/mL, Sigma-Aldrich, St. Louis, MO, USA). On Day 1, the induction medium was supplemented with puromycin (2 µg/mL) for 24 h for the selection of transduced cells. On Day 2, surviving cells were passaged on Matrigel-coated 6-well plates at a concentration of 500 K cells/well in neural differentiation medium (Neurobasal medium (Gibco, Grand Island, NY, USA), B27 supplement (Gibco, Grand Island, NY, USA), Glutamax (1% *v*/*v*, Gibco, Grand Island, NY, USA), Penicillin-Streptomycin (Pen/Strep, 5000 units/mL and 5000 µg/mL respectively, Gibco, Grand Island, NY, USA), D-Glucose (Sigma-Aldrich, St. Louis, MO, USA), BDNF (10 ng/mL, Peprotech, Cranbury, NJ, USA), NT3 (10 ng/mL, Peprotech, Cranbury, NJ, USA), Laminin (200 ng/mL, Millipore Sigma, St. Louis, MO, USA), and Doxycycline (2 µg/mL, Sigma-Aldrich, St. Louis, MO, USA). On Day 4, 50% of the medium was replaced with Neural maturation medium (Neurobasal medium A (Gibco, Grand Island, NY, USA), B27 (Gibco, Grand Island, NY, USA), Glutamax (1% *v*/*v*, Gibco, Grand Island, NY, USA), Pen/Strep, Glucose Pyruvate mix (1:100, final concentration of 5 mM glucose and 10 mM sodium pyruvate), BDNF (10 ng/mL, Peprotech, Cranbury, NJ, USA), NT3 (10 ng/mL, Peprotech, Cranbury, NJ, USA), Laminin (200 ng/mL, Millipore Sigma, St. Louis, MO, USA), and Doxycycline (2 µg/mL, Sigma-Aldrich, St. Louis, MO, USA) supplemented with 4 µM of Cytosineβ-D-arabinofuranoside hydrochloride (Ara-C, Sigma-Aldrich, St. Louis, MO, USA) to inhibit the non-neuronal cell proliferation. From Day 6, 70% of neuronal maturation medium was changed every other day until Day 12. From Day 15, non-doxycycline supplemented maturation medium was used to replace 50% of the medium every 48 h thereafter until Day 21. Mature neuronal cells were collected on Day 21 for RNA isolation and transcriptomic analysis.

### 2.5. Immunocytochemistry Staining for Pluripotency

Immunocytochemistry staining was performed to confirm the pluripotency by using nucleus markers (OCT4 and NANOG) and TRA-1-60 antibodies were used for surface markers. We cultured the iPSCs in the 12-well plates (60,000 cells/well) for 3 days, then cells were quickly washed with PBS and 4% PFA (paraformaldehyde) was used for fixation up to 20 min in 4 °C. Cells were washed with PBS and incubated with blocking buffer with 10% goat serum for 1 h. Primary antibodies OCT4 (#SC-9081), NANOG (#BD-560482), and TRA-1-60 (#MAB4360) were used for staining overnight at 4 °C. Next day cells were washed with PBS and incubated at 37 °C with secondary antibodies and counterstained with DAPI for 20 min at 4 °C.

### 2.6. Flow Cytometry

In order to validate the pluripotency of the cells, the flow cytometry analysis was completed on a BD LSR Fortessa Analyzer (BD Biosciences, Franklin Lakes, NJ, USA) and data were analyzed by using flowJo software (v10.8.1). Induced PSCs were dissociated into single cells with TrypLE (Gibco, Grand Island, NY, USA) and washed with BD-FACS staining buffer (Thermofischer, #00422257, Carlsbad, CA, USA). Then cells were resuspended in the BD-FACS staining buffer and labelled through anti-human TRA-1-60 antibody (Millipore Sigma, St. Louis, MO, USA) and anti-mouse IgM control, PE conjugated (#IC015P), and incubated at 4 °C for 40 min. After incubation, 2 mL of PBS was added in the labelled cells and centrifuged for 5 min at 1500 RPM. Finally, a 200 uL FACS-buffer was used to resuspend the cell and analyzed in BD-Fortessa analyzer (BD, Biosciences, San Jose, CA, USA).

### 2.7. RNA Extraction and Quality Control

Total RNA was isolated from neuronal cell pellets using the Quick-RNA MiniPrep Kit (Zymo Research #R1054) according to the manufacturer’s protocol. Total RNA was quantified using Nanodrop (Thermo Scientific, Waltham, MA, USA). Total RNA of final concentration (400 ng) was aliquoted and shipped to Novogene (Novogene Corporation Inc., Sacramento, CA, USA) for 150 bp paired-end RNA sequencing. Once all the samples passed the primary quality control, library preparation was initiated. See the previously published paper from our group for the descriptive methodology of RNA sequencing and analysis [26,27].

### 2.8. RNA Sequencing and Data Analysis

Next-generation sequencing was outsourced to Novogene Corporation Inc. (Sacramento, CA, USA). The 12 libraries passed Novogene’s quality control and were sequenced in one batch. The company returned to us .fastq files including on average 44.9 million reads per sample with a maximum of 52.7 million and a minimum of 39.8 million. In order to analyze the samples, the 150 bp paired-end reads were aligned to the human reference genome GRCH38 using the package Hisat2 (version 2.2.1) [28], then SAMtools (version 1.1.4) [29] was utilized to produce the corresponding BAM files, and stringtie (version 2.1.7) [30] was used to assemble RNAseq alignments into potential transcripts and estimate their abundance according to GRCh38 human genome annotations [31]. Subsequently, raw counts were computed via the Bioconductor package tximport [32]. Only transcripts with at least 10 reads across all samples were considered for further analysis. The Bioconductor package DESeq2 was used for deferential gene expression analysis [33] and adjusted p (Padj) was calculated using the Benjamini–Hochberg adjustment.

### 2.9. Bioinformatics Analysis

To identify possible outliers due to experimental conditions, we performed principal component analysis using read counts as input and the ggplot2 and ggfortify R packages. For UMAP analysis, we used the R functions prcomp and umap.

To validate the results through comparison with other datasets, the differentially expressed genes (DEG) were filtered by the adjusted *p*-values by DEseq2 (FDR 0.05, 0.1 and 0.2 for our dataset) and intersected with the data reported by Mathys et al. [34] for excitatory cortical neurons (FDR between 5 × 10^−5^, 5 × 10^−6^, and 5 × 10^−7^ for the Mathys dataset due to the larger number of positives).

To confirm that the DEGs were a valid gene set that could potentially indicate functional differences between AD and control-derived neurons, we utilized the STRING database of protein–protein interactions [35] version 11.5 (https://string-db.org/, accessed on 3 May 2023) to determine whether these genes exhibited more frequent interactions than predicted by chance. Input genes were those with adjusted *p*-value < 0.05 of which 485 were recognized by STRING. The default parameters were used in the STRING interface, including a medium interaction confidence (0.4) and restricting to only the query genes without an additional level of interactors, as suggested by the authors for meaningful edge (interaction) enrichment results.

## 3. Results

### 3.1. Generation of iPSCs and Excitatory Neurons

Human iPSCs were generated through PMNCs acquired from the Johns Hopkins Core facility. Cells were reprogrammed through the transient expression method by using episomal vectors. Nucleofector cells were grown in the culture for a minimum of two weeks to develop iPSC-like colonies [12,23]. The morphological identity of the iPSCs was captured by brightfield microscopy (Figure 1a). The immunofluorescence staining with nucleic markers (NANOG and OCT4) and surface marker (TRA-1-60) confirmed pluripotency which was also validated by flow cytometry for TRA-1-60 positive cells (Figure 1c). The functional pluripotency of the iPSCs was assessed by the in vitro trilineage differentiation into three germ layers as previously reported [22]. We did not observe any chromosomal abnormality by karyotyping.

To assess the excitatory neuronal identity of the cells produced by Ngn2 induction, we used the RNA sequencing results to assess the expression of the expected marker genes. A list of genes and their expression levels is shown in Table 2. We note that we, as well as others, have used this protocol repeatedly and consistently to obtain cells with transcription profiles resembling those of excitatory neurons [17,25,26,27,36,37]. For comparison, we provide Figure 2, a heatmap for the expression of these genes in human excitatory neurons from the single-cell expression database available by the Allen Brain Atlas (https://celltypes.brain-map.org/rnaseq/human_m1_10x, accessed on 26 July 2023).

### 3.2. RNA Sequencing

We received an average of 45.1 million paired reads per sample (min = 39.8, max = 52.7) with an error rate of 0.03%, of which at least 96.3% had a Phred score of 20 and at least 90.1 of those had a Phred score of 30. Principal component analysis (PCA) using all genes and the Fragments Per Kbp per million (FPKM) showed that sample AD06 was an outlier, likely the result of unknown differences during cell growth and differentiation, and it was removed from further analysis (Figure 3). The remaining samples were then used for differential expression (DE) analysis. This analysis identified 621/22,011 DE genes at Padj < 0.05, 204 higher in AD, and 417 lower.

### 3.3. Bioinformatics Analysis

The main question of this project is whether the transcriptome of neuronal cells derived from AD patients shows detectable differences as compared to those derived from the controls, making them appropriate for the study of the disease and potentially the identification of biomarkers. Having identified a significant number of genes changing expression, we set out to test whether these genes are a meaningful set and related to AD rather than a collection of random genes from experimental noise. Therefore, we asked three specific questions: Are the identified genes related to each other? Do they overlap with genes identified by comparable in vivo studies? Do they contain genes that are known to be involved in Alzheimer’s disease? Lastly, we investigated whether the transcriptome of neurons derived from patients and controls could potentially be used for meaningful classification of individuals to facilitate precision medicine.

### 3.4. Dysregulated Genes Are Part of a Gene Network

To explore whether the observed set of dysregulated genes is an apparently random collection or a functionally meaningful set, we queried the STRING database which systematically collects and integrates physical interactions and functional associations. Additionally, for the interactive visualization of these interactions in a provided gene set, it allows the assessment of excess of edges (connections) between the nodes (proteins/genes) compared to a random set. Of the 485 genes corresponding to known proteins at Padj < 0.05, there were 801 interactions, compared to an expected 662 (p-enrichment 9 × 10^−8^). The genes corresponding to the interacting proteins were enriched for those expressed in the brain (FDR = 3 × 10^−5^).

### 3.5. Validation through Comparison with External Data

As opposed to most transcriptomic studies of AD that examine bulk tissue from the brain, our data come from a more homogeneous population of cells after reprogramming and differentiation to excitatory neurons. We sought an appropriate dataset to examine the validity of our results by comparing the identified differentially regulated genes. No published dataset fully matched our conditions, yet one recent study performed single-cell sequencing providing data specific to excitatory neurons from the prefrontal cortex of 48 individuals with varying degrees of AD pathology (Mathys et al. [34] from their Supplementary Table 3, tab Ex, no pathology vs. pathology). Due to our small sample size, we examined genes with a corrected *p*-value of <0.05, 0.1, and 0.2 in our study, while we examined *p*-value thresholds <5 × 10^−5^, 5 × 10^−6^, and 5 × 10^−7^, for the Mathys et al. [34] study which reported many more positive results (Table 3). We consistently found a significant excess of overlap from that expected by chance (lowest *p* = 3.5 × 10^−5^). We also consistently observed significantly more concordance in the direction of change than that expected by chance (Table 3), reaching a binomial test *p*-value of 3.2 × 10^−4^. The enrichment exceeded 1.5-fold and the concordance rate was 75% for some thresholds. These results suggest that many of the genes we found to have expression differences mirror changes observed in disease despite our small sample size and despite coming from blood cells that have been reprogrammed and undergone differentiation. It further suggests that the genes driving this overlap represent primary expression changes related to disease risk and not secondary changes due to the disease itself and/or the prescribed medications.

Many genes related to AD were among the differentially expressed (DE) genes in AD patients versus the control-derived samples. A complete list of genes with *p*-values is provided in Appendix A. Notably, among genes at *p*-adjusted <0.05 was APOE that was expressed less in patient-derived neurons, with an additional four genes identified by large GWAS for AD: *CD2AP*, *RBFOX1*, *TMEM132C*, and *NPAS2*. A complete list of genes with all normalized counts in excitatory neurons (RNAseq) for all individuals included in the analysis is also provided in Appendix A.

### 3.6. Use of Transcriptome Data as a Patient Classification Tool

Having shown that iPSC-derived neurons from individuals with AD show transcriptome differences from controls, we performed exploratory analysis to determine whether the transcriptome of such cells may also be used to cluster affected individuals in groups that might be helpful in determining the course and/or best treatment options, a major goal of precision medicine. Given our small sample size, we did not expect statistical support, but perhaps an indication of whether this might be a useful path going forward. To determine whether the AD patient transcriptome could be used as a multidimensional variable for patient classification, we used Uniform Manifold Approximation and Projection for Dimension Reduction (UMAP), a technique that can be used to visualize patterns of clustering in high-dimensional data [38] that is frequently used for the analysis of single-cell data. Figure 4 shows the results of this analysis. We observed that the seven patients appeared to form two clusters on the UMAP1 axis, one containing patients 1, 3, 5, and 8 and the other containing patients 2, 4, and 7. We compared cognitive data between clusters for the individuals with detailed clinical data (all but AD_06 and AD_08). Interestingly, despite having similar clinical dementia ratings (CDR) in the CDR scale [39], the two groups were significantly different in age at examination but despite that the younger group, had consistently lower performance in letter fluency [40] “Number of words” test and on the Mini-Mental State Examination (MMSE) [41] (Table 4). These differences did not reach statistical significance, as would be expected by the very small numbers. Moreover, while the three male patients do cluster together, AD_01 who is female, also clusters in the middle of them, making it unlikely that this is a sex effect. Given that the patient sample alone is small, this is not unlikely to be by chance.

## 4. Discussion

We generated iPSCs from 14 individuals and after differentiation to excitatory neurons, we found that the patients with AD and the controls showed significant transcriptomic differences. The genes involved appear functionally connected to each other and overlap with genes identified in prior in vivo analysis by another research group, supporting the validity of our results. This is an important observation as it supports the use of measurements from patient-derived cells differentiated into neurons for the identification of biomarkers for AD. Further, our exploratory patient-only analysis of clustering suggested two possible clusters, indicating it might be possible to group patients into biologically meaningful groups that may advance precision medicine.

The genes differing between the cases and controls included APOE, a very well-established AD risk gene that was found to be expressed lower in patient-derived cells, and others previously linked to AD (*CD2AP*, *RBFOX1*, *TMEM132C*, *NPAS2*). *CD2AP*, whose levels were decreased in the patient-derived cells, is a scaffolding molecule reported to be associated with AD [42]. Its mRNA levels have also been found to be decreased in peripheral lymphocytes of sporadic AD patients and *CD2AP* loss of function has been linked to enhanced Aβ production, Tau-induced neurotoxicity, abnormal neurite structure modulation, and reduced blood–brain barrier integrity [43]. *RBFOX1*, whose levels were increased in the patient derived cells is another AD associated gene [44], linked to multiple additional psychiatric traits [45]. It is an RNA-binding protein that regulates alternative splicing [46] including that of APP [47] and variants near it also regulate gene co-expression modules in the aging human brain [48]. *TMEM132C*, whose levels were increased in the patient-derived cells, encodes a neural adhesion molecule associated with AD [44] which is also independently associated with cognitive impairment in a hypotensive population [49] and with high-altitude adaptation in Tibetans. *NPAS2*, whose levels were decreased in the patient-derived cells, is a circadian rhythm gene, a system that has been linked to many diseases including dementia [50]. Interestingly, *NPAS2* has also been implicated in prion diseases [51,52] and is a regulator of genes controlling inflammation in AD [53].

The highest ranked genes in our study, *HS6ST2* and *FBXO2*, have also been previously linked to AD. *HS6ST2*, along with its isoform *HS3ST2*, have been found to increase in AD and *HS3ST2*, which through further study, was found to be critical for the abnormal phosphorylation of tau [54]. Most notably, *FBXO2* has been found to regulate amyloid precursor protein levels and processing [55] and alter cognitive behavior in mice [56].

While there seem to be biologically relevant differences between the transcriptomes of iPSC-derived neurons from patients and controls, our sample is too small to reach conclusions about the possible classification of patients based on these transcriptomes. Our exploratory analysis using UMAP, a tool for visualizing the proximity of different samples in a multidimensional space (transcriptome) in two dimensions, suggests two possible clusters. Interestingly the clusters showed some clinical differences, with one cluster (AD_2, and 4) having a lower age at presentation, showing similar clinical dementia ratings, but showing potentially worse scores in MMSE and the “Number of words” test. We expect that with larger samples more of these differences will emerge that might reflect patient subgroups and determine the usefulness of this clustering approach for precision medicine. Incorporated in our mainstream analysis of patients for precision medicine, this could become a powerful tool. Despite the small sample size, our positive results indicate that the differences we observed are likely only the tip of the iceberg of information we might achieve with large patient samples when studying iPSC-derived excitatory neurons. These preliminary findings suggest that this methodology has the potential for better understanding AD mechanisms and identifying potential targets for treatment, as well as better understanding the heterogeneity of mechanisms within AD.

## Figures and Tables

**Figure 1 cells-12-01990-f001:**
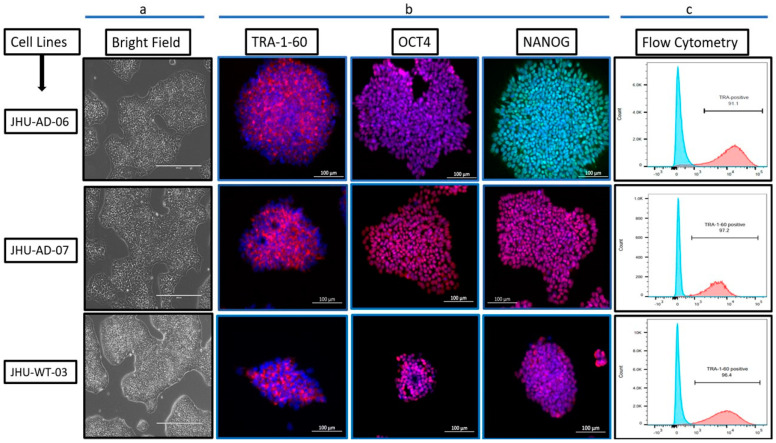
Characterization of iPSCs: (**a**) Brightfield microscope shows the iPSC morphology (Scale bars: 400 μm). (**b**) Fluorescent immunostaining by using pluripotency markers (TRA-1-60, OCT4, Nanog) (Scale bars: 100 μm). (**c**) Flow-cytometry shows the purity of the iPSCs by pluripotency marker (TRA-1-60^+^ cells).

**Figure 2 cells-12-01990-f002:**
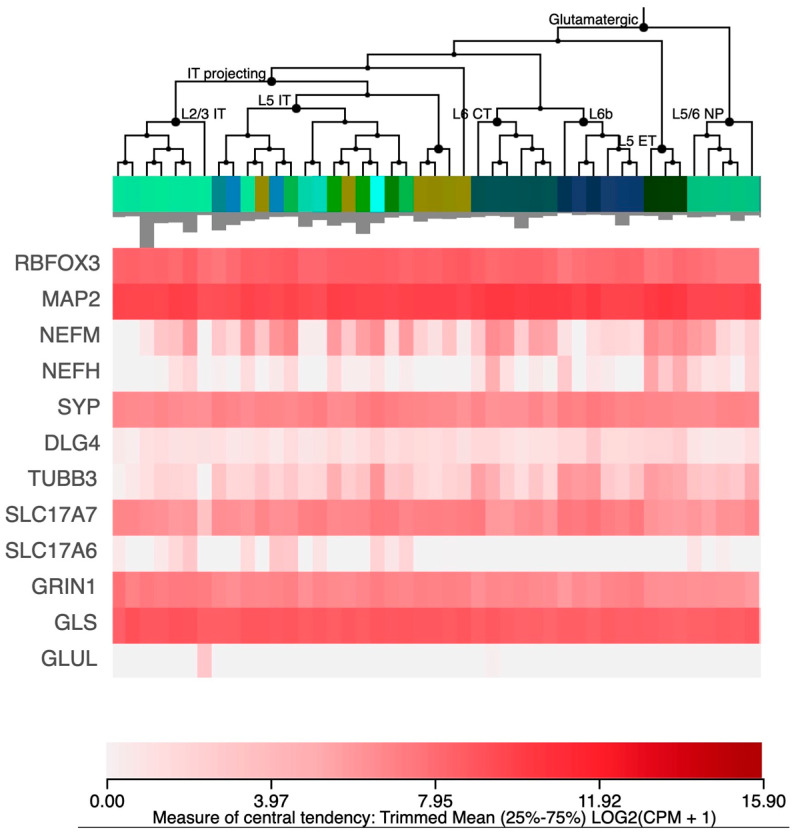
Expression of marker genes in human brain. Heatmap of the expression of the genes in Table 2 in human excitatory neurons from the single-cell expression database available from the Allen Brain Atlas. For details on subtypes and single cells, see the Allen Brain Atlas database.

**Figure 3 cells-12-01990-f003:**
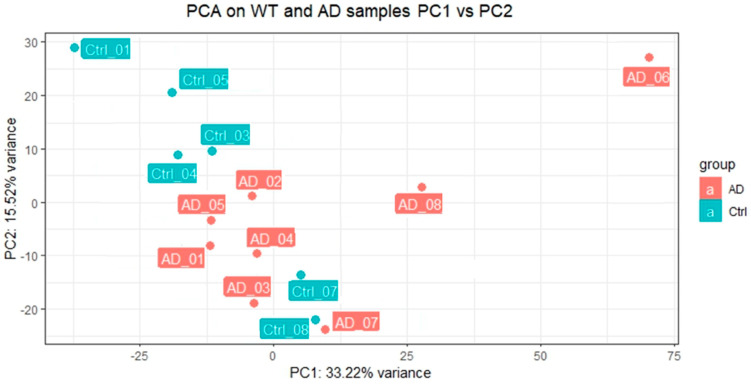
Principal component analysis (PCA) using all genes on AD and control samples.

**Figure 4 cells-12-01990-f004:**
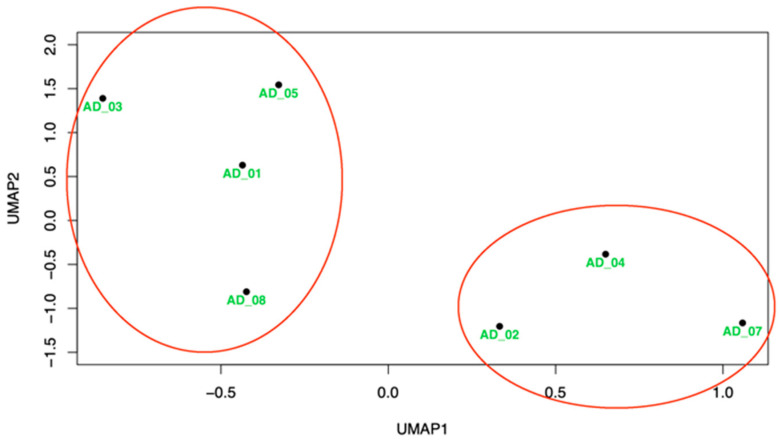
Identification of AD patients’ clusters by UMAP.

**Table 1 cells-12-01990-t001:** Diagnosis and demographics of patients and healthy individuals used in the transcriptomic studi-s.

Tracking ID	Phenotype	Age	Gender	Source of iPSC
JHU-AD-01	AD	68	F	PBMC
JHU-AD-02	AD	60	F	PBMC
JHU-AD-03	AD	70	M	PBMC
JHU-AD-04	AD	62	F	PBMC
JHU-AD-05	AD	72	M	PBMC
JHU-AD-06	AD	55	F	PBMC
JHU-AD-07	AD	89	F	PBMC
JHU-AD-08	AD	81	M	PBMC
JHU-WT-01	Normal	56	F	Skin Fibroblast
JHU-WT-03	Normal	81	F	PBMC
JHU-WT-04	Normal	85	F	PBMC
JHU-WT-05	Normal	56	F	Skin Fibroblast
JHU-WT-07	Normal	88	F	PBMC
JHU-WT-08	Normal	84	F	PBMC

**Table 2 cells-12-01990-t002:** Expression of excitatory neuronal markers in iPSC-derived neurons.

Marker	Gene Name	Identity	PFKM
NeuN	RBFOX3	Neuronal	22.5
MAP2	MAP2	Neuronal	197
160 kDa neurofilament Medium	NEFM	Neuronal	2428
200 kDa neurofilament Heavy	NEFH	Neuronal	6.5
Synaptophysin	SYP	Neuronal	207
PSD95	DLG4	Neuronal	134
TUJ1	TUBB3	Neuronal	1384
vGluT1	SLC17A7	Glutamatergic	24
vGluT2	SLC17A6	Glutamatergic	107
NMDAR1	GRIN1	Glutamatergic	5.4
Glutaminase	GLS	Glutamatergic	29.3
Glutamine synthetase	GLUL	Glutamatergic	27

**Table 3 cells-12-01990-t003:** Comparison of identified differentially regulated genes with external data.

SC Study Threshold p (N Genes)	Our Study Threshold p (N Genes)	Overlap	Fold Excess over Expected	𝓧 Test *p* Value	Concordance %	Concordance N	Binomial *p*-Value
0.00005 (1331)	0.05 (394)	46	1.060	n.s.	72%	33	0.002
0.000005 (899)	0.05 (394)	36	1.228	n.s.	75%	27	0.002
0.00005 (1331)	0.1 (765)	109	1.293	0.003196507	60%	65	0.0275
0.000005 (899)	0.1 (765)	86	1.511	0.0000351055	62%	53	0.02
0.00005 (1331)	0.2 (1547)	200	1.173	0.010186373	59%	118	0.0065
0.000005 (899)	0.2 (1547)	146	1.268	0.001361469	64%	94	0.00032
0.0000005 (632)	0.2 (1547)	100	1.236	0.01974401	63%	63	0.006

**Table 4 cells-12-01990-t004:** Clinical differences of the identified patient clusters. Sum of FAS is the sum of number of words starting from F, A, and S. MMSE is the score on the MMSE including serial 7s.

Patient	Age-at-Examination	Sum of FAS	MMSE	Overall CDR Rating	CDR Sum of Boxes
AD-4	62	17	18	1	4.5
AD-2	60	23	17	0.5	3.5
AD-1	68	48	24	0.5	2.5
AD-3	70	47	25	0.5	3
AD-5	72	27	19	1	8.5
*t*-test *p*	0.01	0.108	0.123	0.789	0.81

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
