# Peer review of "Excitatory Neurons Derived from Human-Induced Pluripotent Stem Cells Show Transcriptomic Differences in Alzheimer’s Patients from Controls"

_cells, 2023, doi:10.3390/cells12151990_

Round 1
Reviewer 1 Report
In the manuscript “Excitatory neurons derived from human induced pluripotent stem cells show transcriptomic differences in Alzheimer’s patients from controls”, Sagar et al investigate transcriptomic differences between iPSC-derived neurons derived from 8 AD patients of mixed gender and 6 cognitively unimpaired controls of female gender and of similar age to the AD cases.
The authors obtained samples of adult somatic cells, peripheral blood mononuclear cells for AD patients and skin fibroblasts for normal controls. These cells were first reprogrammed into iPSCs and then differentiated into excitatory glutamergic neurons - these were subjected to transcriptomic analysis and clustering.
They find that 621 genes show differences in expression levels between the AD cases and control derived neurons. These genes show significant overlap and general concordance with genes reported from a single cell transcriptomic study of AD patients, including five genes implicated in AD by genome wide association studies and appear to be part of a larger functional network. Exploratory analysis with Uniform Manifold Approximation and Projection suggests distinct clustering of patients based on gene expression who may be clinically different.
Even though the sample size is small, the results are very intriguing and raise the possibility that this approach could potentially be used to subgroup AD.
This is potentially a very interesting study but a concern about this manuscript lies with the vagueness of the described workflow and the lack of quality controls. It is understood from lines 76-84 of the manuscript that AD iPSCs were derived from peripheral blood mononuclear cells obtained from the blood samples and the two control-derived iPSCs (Ctrl 1 and Ctrl 5, both of the same age and gender) were obtained from adult skin fibroblasts. Nowhere in the methods is it explained how the fibroblasts were reprogrammed and if the same protocol used? This is not my area of expertise but is it known that iPSC derived from different adult somatic cells by different procedures are nevertheless identical?
Any small differences due to the somatic identity or reprogramming procedure would influence the results and the conclusions drawn. It is stated in the methods that the iPSCs were differentiated into glutamergic excitatory neurons; do the neurons achieve the same differentiation state. Did the authors purify neurons or did they extract RNA from a mixed culture. Any variation in the differentiation state or the proportion of differentiated cells would also affect the transcriptomic results and the conclusions.
Another issue is the potential variability of the differentiated cultures? Also how reproducible is the transcriptomic data if the differentiation is undertaken multiple times?
According to literature, multiple factors such as cell type of origin, gender of the donor, culture conditions, and reprogramming method contribute to variation between iPSC lines, so it is essential that control and disease cases are comparable by design and the workflow is communicated clearly. Lack of rigorous quality control at the reprogramming stage could, among other factors, may explain an outlier AD_06 observed at the level of neuronal genes in Fig.2. Also, curiously, in AD patients clusters shown in Fig. 3, all male-derived samples cluster together – can this be the influenced by gene differences of iPSCs derived from male donors that carried over into iPSC-derived neurons?
Minor remarks:
Fig.1c: panels are unreadable
Line 27: “finding will to contribute” please check sentence
Line 32: “Lacking a clinically effective treatment for AD…” - what about the newly approved lecanemab (leqembi), should not it be mentioned?
Author Response
Reviewer #1: In the manuscript “Excitatory neurons derived from human induced pluripotent stem cells show transcriptomic differences in Alzheimer’s patients from controls”, Sagar et al investigate transcriptomic differences between iPSC-derived neurons derived from 8 AD patients of mixed gender and 6 cognitively unimpaired controls of female gender and of similar age to the AD cases. The authors obtained samples of adult somatic cells, peripheral blood mononuclear cells for AD patients and skin fibroblasts for normal controls. These cells were first reprogrammed into iPSCs and then differentiated into excitatory glutamergic neurons - these were subjected to transcriptomic analysis and clustering. They find that 621 genes show differences in expression levels between the AD cases and control derived neurons. These genes show significant overlap and general concordance with genes reported from a single cell transcriptomic study of AD patients, including five genes implicated in AD by genome wide association studies and appear to be part of a larger functional network. Exploratory analysis with Uniform Manifold Approximation and Projection suggests distinct clustering of patients based on gene expression who may be clinically different. Even though the sample size is small, the results are very intriguing and raise the possibility that this approach could potentially be used to subgroup AD.
¨ This is potentially a very interesting study but a concern about this manuscript lies with the vagueness of the described workflow and the lack of quality controls. It is understood from lines 76-84 of the manuscript that AD iPSCs were derived from peripheral blood mononuclear cells obtained from the blood samples and the two control-derived iPSCs (Ctrl 1 and Ctrl 5, both of the same age and gender) were obtained from adult skin fibroblasts. Nowhere in the methods is it explained how the fibroblasts were reprogrammed and if the same protocol used? This is not my area of expertise but is it known that iPSC derived from different adult somatic cells by different procedures are nevertheless identical?
We appreciate the Reviewer’s comment. We haven’t given any details on how the fibroblasts were reprogrammed because the method has been reported previously (Mahairaki et al. 2014; Ref. 20 in the submitted Manuscript). There we describe the generation and extensive characterization of the iPSC lines used in this study. We used the same method or reprogramming for both the PBMCs and the fibroblasts, which is nucleofection with nonintegrating episomal three plasmid vectors.
¨ Any small differences due to the somatic identity or reprogramming procedure would influence the results and the conclusions drawn. It is stated in the methods that the iPSCs were differentiated into glutamergic excitatory neurons; do the neurons achieve the same differentiation state. Did the authors purify neurons or did they extract RNA from a mixed culture. Any variation in the differentiation state or the proportion of differentiated cells would also affect the transcriptomic results and the conclusions.
We thank the reviewer for the comment. Although the protocol is designed to generate only neurons, some heterogeneity is possible and could reduce power but is unlikely to produce false positives in the absence of a consistent difference between cases and controls. We have addressed heterogeneity in a separate paper in BioRxiv (doi: https://doi.org/10.1101/2022.06.15.495952).
¨ Another issue is the potential variability of the differentiated cultures? Also how reproducible is the transcriptomic data if the differentiation is undertaken multiple times?
Although we did not perform direct comparison for differentially expressed genes due to different experimental designs, we did perform multiple studies with Ngn2 differentiated cells and observe similar neuronal marker expression profiles (PMID: 36672919, 33497908, Feuer et al, molecular psychiatry in press and doi: https://doi.org/10.1101/2022.06.15.495952).
¨ According to literature, multiple factors such as cell type of origin, gender of the donor, culture conditions, and reprogramming method contribute to variation between iPSC lines, so it is essential that control and disease cases are comparable by design and the workflow is communicated clearly. Lack of rigorous quality control at the reprogramming stage could, among other factors, may explain an outlier AD_06 observed at the level of neuronal genes in Fig.2. Also, curiously, in AD patients clusters shown in Fig. 3, all male-derived samples cluster together – can this be the influenced by gene differences of iPSCs derived from male donors that carried over into iPSC-derived neurons?
While the three male patients cluster together, AD_01 who is female also clusters in the middle of this group making it unlikely that this is a sex effect. Given that the patient sample alone is small this is could have occurred by chance. Nevertheless, we recognize this is a possible limitation and mention it in the Results. We have also taken care to explicitly emphasize that this analysis is exploratory, serving as a pilot for future confirmatory studies involving larger sample sizes.
¨Minor remarks:
Fig.1c: panels are unreadable
Following the reviewer’s suggestions, we improved the quality of the Fig.1c and have submitted a revised one.
Line 27: “finding will to contribute” please check sentence
We have rephrased the sentence to make it clear as follows:
“Our research outcomes will enable precise identification of distinct biological subtypes among individuals with Alzheimer's disease, facilitating the implementation of tailored precision medicine strategies.”
Line 32: “Lacking a clinically effective treatment for AD…” - what about the newly approved lecanemab (leqembi), should not it be mentioned?
We thank the reviewer for the comment. We believe that even with lecanemab and donanemab there is no clinically effective treatment of AD. The newly approved drug lecanemab only minimally delays progression of AD symptoms, and does not fully treat the disease. This is acknowledged widely by the companies that have produced these drugs. Nevertheless, we have slightly modified the wording to be less confident. This now reads: “lacking disease therapies that reverse or substantially delay progression of symptoms.”
Reviewer 2 Report
The manuscript describes a comparison of iPSC-derived excitatory neurons from eight AD patients and six neurologically normal controls. The Authors used RNA-seq to explore transcriptomic differences due to disease status and found 621 differentially expressed genes (DEG) that displayed enrichment for brain proteins and substantial overlap to genes differentially expressed in excitatory neurons from a post-mortem brain snRNA-seq dataset (Mathys et al.).
The Authors were motivated to use the latter dataset because there were no ‘IPSC-derived data available. However there have been quite a few descriptions of excitatory neurons from IPSCs in the literature e.g. reviews (PMID: 31391546 and 35107767). Similarly, some are transcriptomic studies e.g. PMID: 34921933 and 30699343. Is the point being made by the Authors that there have been no previous transcriptomic studies of iPSC-derived excitatory neurons from AD patients? Otherwise, it is not clear why other iPSC-derived neuronal data could not be compared here and, or why no comparisons with previous data sets are made in the Discussion. Importantly, is this study completely novel?
The appropriateness of the comparison with post-mortem brain derived single nuclei data also needs to be given additional consideration by the Authors. As correctly described here, the iPSC process removes epigenetic marks from the antecedent cells (a combination of PMBCs and skin fibroblasts) allowing ‘baseline’ genetic differences in gene expression to be explored in subsequent neurons rather than effects of the disease. However, post-mortem brain cells from AD patients will have such epigenetic marks resulting from ageing, AD and potentially co-morbidities. Therefore, is the overlap seen here more than might be expected? Can the Authors describe where the think the data is likely to be more, or less concordant?
The Authors review five DEGs here that are also previously linked to AD via GWAS studies. These genes are ranked from 64th, 133rd, 244th, 370th and 419th among the 621 DEGs in the Supplementary Table? What about the highest ranked genes such as HS6ST2 and FBXO2, the latter being most highly expressed in the brain?
In the Methods, the Authors describe one AD patient (07) from an alternative study but also two controls (01 and 05) from skin fibroblasts. The latter, in particular, are not mentioned further when describing the derivation of neurons (see below) or possible reasons for outliers e.g. Figure 2. Where in fact 06 seems to be the most disparate from the other cell lines.
Minor issues
Line 36 – Add PMID: 35379992 who describe 75 loci now from GWAS studies.
Line 38 – Does the recent success of drugs like lecanemab mean this point might need nuancing?
Line 69 – there have been previous studies producing glutaminergic neurons via iPSCs cells so please describe how the current study is novel (ie) motivation for the work.
Line 83 – from from
Line 92 – Table 1 – consider adding column for source of iPSC (skin or PBNCs)
Line 97 – Skin fibroblasts?
Line 148 – upto
Line 151 – Did the Authors consider IHC for the differentiated neurons as per iPSCs.
Line 153 – Flow cytometry
Line 164 – Total RNA was isolated from neuronal cell pellets using….
Line 224 – Table 2. Consider a column comparing with another study. N.B Using the Allen Brain Single Cell data, GLUL is mainly in astrocytes while GLS is pan neuronal.
Line 256 – Can enriched pathways be added here to show proteins are “functioning in the brain”.
Line 259 – bulk tissue?
There are a few typos as noted above
Author Response
Reviewer #2 :
The manuscript describes a comparison of iPSC-derived excitatory neurons from eight AD patients and six neurologically normal controls. The Authors used RNA-seq to explore transcriptomic differences due to disease status and found 621 differentially expressed genes (DEG) that displayed enrichment for brain proteins and substantial overlap to genes differentially expressed in excitatory neurons from a post-mortem brain snRNA-seq dataset (Mathys et al.).
¨The Authors were motivated to use the latter dataset because there were no ‘IPSC-derived data available. However there have been quite a few descriptions of excitatory neurons from IPSCs in the literature e.g. reviews (PMID: 31391546 and 35107767). Similarly, some are transcriptomic studies e.g. PMID: 34921933 and 30699343. Is the point being made by the Authors that there have been no previous transcriptomic studies of iPSC-derived excitatory neurons from AD patients? Otherwise, it is not clear why other iPSC-derived neuronal data could not be compared here and, or why no comparisons with previous data sets are made in the Discussion. Importantly, is this study completely novel?
We would like to thank the reviewer for comments. The transcriptomic studies mentioned are very small (PMID: 34921933 study of only 8 individuals total, and PMID: 30699343 of only 11 individuals) and using different differentiation protocols which is unlikely to have power to show significant overlaps with our also small, yet twice the size study. We chose the largest available similar dataset. Many aspects of our study, especially the clustering analysis, are unique and an important pilot for future work.
¨The appropriateness of the comparison with post-mortem brain derived single nuclei data also needs to be given additional consideration by the Authors. As correctly described here, the iPSC process removes epigenetic marks from the antecedent cells (a combination of PMBCs and skin fibroblasts) allowing ‘baseline’ genetic differences in gene expression to be explored in subsequent neurons rather than effects of the disease. However, post-mortem brain cells from AD patients will have such epigenetic marks resulting from ageing, AD and potentially co-morbidities. Therefore, is the overlap seen here more than might be expected? Can the Authors describe where the think the data is likely to be more, or less concordant?
We would like to thank the reviewer for the comments. The epigenetic marks are erased in iPSCs and reintroduced during neuronal differentiation. We expect that the marks characterizing excitatory neurons will be the same in both AD and controls but the marks due to aging or disease will not. As a result, we think that the overlapping Differentially Expressed (DE) genes are those that are different due to the genetic predisposition – not the disease or aging process, which makes the information from iPSC-derived neurons a valuable addition shedding light to early origins of disease.
¨The Authors review five DEGs here that are also previously linked to AD via GWAS studies. These genes are ranked from 64th, 133rd, 244th, 370th and 419th among the 621 DEGs in the Supplementary Table? What about the highest ranked genes such as HS6ST2 and FBXO2, the latter being most highly expressed in the brain?
We thank the reviewer for the comments. We have added the following paragraph in the Discussion of the revised manuscript: “ The highest ranked genes in our study HS6ST2 and FBXO2, have also been previously linked to AD. HS6ST2, along with its isoform HS3ST2 have been found increase in AD and HS3ST2 which was further studied was found critical for the abnormal phosphorylation of tau [PMID: 25842390]. Most notably FBXO2 has been found to regulate amyloid precursor protein levels and processing [PMID: 24469452] and alter cognitive behavior in mice [PMID: 36609445].
¨In the Methods, the Authors describe one AD patient (07) from an alternative study but also two controls (01 and 05) from skin fibroblasts. The latter, in particular, are not mentioned further when describing the derivation of neurons (see below) or possible reasons for outliers e.g. Figure 2. Where in fact 06 seems to be the most disparate from the other cell lines.
As shown in Figure 2 the origin of the cells does not seem to make a difference in their transcriptome as they cluster together with all the other cells. The only outlier, neurons from patient AD_06 derived , were from blood mononuclear cells yet do not cluster with the rest similarly derived neurons. We have now added this observation in the Results of the revised manuscript.
¨Minor issues
Line 36 – Add PMID: 35379992 who describe 75 loci now from GWAS studies.
We have added the recommended citation in the manuscript and on the reference list.
Line 38 – Does the recent success of drugs like lecanemab mean this point might need nuancing?
We thank the reviewer for the comment. We believe that even with lecanemab and donanemab there is no clinically effective treatment of AD. The newly approved drug lecanemab only minimally delays progression of AD symptoms, and does not fully treat the disease. This is acknowledged widely by the companies that have produced these drugs.
We have added a note in the Introduction of the revised manuscript saying “Even with the newly approved drugs (lecanemab and donanemab) there is no clinically effective treatment of AD. Lecanemab only minimally delays progression of AD symptoms, and does not fully treat the disease. The newly approved drug lecanemab only minimally delays progression of AD symptoms, and does not fully treat the disease.”
Line 69 – there have been previous studies producing glutaminergic neurons via iPSCs cells so please describe how the current study is novel (ie) motivation for the work.
We would like to thank the reviewer for the comment. The novelty in our studies is that we are having a larger sample size that we used it to cluster patients by transcriptome.
Line 83 – from from
The duplication has been corrected in the Revised manuscript.
Line 92 – Table 1 – consider adding column for source of iPSC (skin or PBNCs)
We thank the reviewer for the comment. We have attached a revised table according to the reviewer’s recommendation.
Line 97 – Skin fibroblasts?
We apologize for the confusion. At the Materials and Methods (Patient Section) we mentioned that “Adult skin fibroblasts from cognitively unimpaired individuals (JHU-WT-01 and -05) were obtained from the Coriell Institute for Medical Research 20 “.
Line 148 – upto
We have made the correction in the revised manuscript.
Line 151 – Did the Authors consider IHC for the differentiated neurons as per iPSCs.
We would like to thank the reviewers for the comment.
We have performed IHC for the iPSCs as part of the standard characterization for pluripotency of generated iPSC lines.
Since the focus of the paper is the transcriptomics analysis we haven’t performed IHC for the neurons.
We have performed multiple studies though with Ngn2 differentiated cells where we have included IHC (PMID: 36672919, 33497908, Feuer et al, molecular psychiatry in press and doi: https://doi.org/10.1101/2022.06.15.495952).
Line 153 – Flow cytometry
We made the correction.
Line 164 – Total RNA was isolated from neuronal cell pellets using….
We thank the reviewer for the suggestion.We did the change in the manuscript.
Line 224 – Table 2. Consider a column comparing with another study. N.B Using the Allen Brain Single Cell data, GLUL is mainly in astrocytes while GLS is pan neuronal.
We thank the reviewer for the suggestion. The Allen Brain Atlas does not provide cell type specific data so the addition would not be meaningful to add to this table which is specific to the induced cells.
Line 256 – Can enriched pathways be added here to show proteins are “functioning in the brain”.
We apologize but we didn’t understand the reviewer’s comment so we haven’t make any further additions or modifications to the revised manuscript.
Line 259 – bulk tissue?
We have made the correction in the revised manuscript.
Reviewer 3 Report
Stem cell technology enables us to generate virtually any human cell type from patient-specific iPSCs. However, reprogramming into pluripotency leaves no traces of their donors’ ages, representing a more significant challenge when investigating age-dependent diseases. In contrast, the direct conversion of human PMNCs into induced neurons preserves substantial signatures of human aging. This is an interesting work, and I have a few questions and suggestions for authors.
1. The author hasn’t mentioned samples are taken from sporadic or Familial AD patient
2. The author must show Ab42/40 ratios to ensure these patients are Familial or sporadic AD donner.
3. Synaptic disturbances in excitatory to inhibitory balance in hippocampus and cortex circuits are thought to contribute to the progression of AD and Dementia. Why does the author think differentiating the iPSC into excitatory neurons will give a better inside of AD pathology?
Some suggestions for authors, if they can put immunofluorescence images quantification and a few RNAseq graphs, will make the manuscript more appealing e.g.
Control vs. AD
1- bIII-tubulin, and NeuN
2- vGlut1/MAP2
3- GABA/bIII-tubulin
4- Pre and postsynaptic marker density
5- AD-specific deferential gene expression in PMNCs (RNAseq)
6- AD-specific deferential gene expression in Excitatory neurons (RNAseq)
7- Spontaneous neuronal activity
Author Response
Reviewer #3 :
Stem cell technology enables us to generate virtually any human cell type from patient-specific iPSCs. However, reprogramming into pluripotency leaves no traces of their donors’ ages, representing a more significant challenge when investigating age-dependent diseases. In contrast, the direct conversion of human PMNCs into induced neurons preserves substantial signatures of human aging. This is an interesting work, and I have a few questions and suggestions for authors.
¨The author hasn’t mentioned samples are taken from sporadic or Familial AD patient
We would like to thank the reviewer for this note. These are sporadic AD patients based on age of onset and no reports of APP, PSEN1 or PSEN2 mutations. We have added the clarification to the revised manuscript at the Patients and Controls session.
¨The author must show Ab42/40 ratios to ensure these patients are
Familial or sporadic AD donner.
We thank the reviewer for his note. The participants have all received a diagnosis of AD (sporadic or familial) confirmed by CSF or imaging biomarker or by autopsy. The Ab42/40 ratio would not add any more information since there is no widely accepted cut off of this ratio that can be used for diagnosis.
¨Synaptic disturbances in excitatory to inhibitory balance in hippocampus and cortex circuits are thought to contribute to the progression of AD and Dementia. Why does the author think differentiating the iPSC into excitatory neurons will give a better inside of AD pathology?
We would like to thank the reviewer for the comment. The balance hypothesis in not clearly proven. There is however good evidence that excitotoxicity is involved in spread of the disease. In fact , one of the approved medications (Memantine) targets Glutamate excitotoxicity. So, we believe that iPSC-derived excitatory neurons will give additional (not better) insights in AD pathology. Since the iPSC models allow the differentiation in a specific cell type (excitatory neurons) at a time, is seems that excitotocity is the best place to start.
¨Some suggestions for authors, if they can put immunofluorescence images quantification and a few RNAseq graphs, will make the manuscript more appealing e.g.
Control vs. AD
1- bIII-tubulin, and NeuN
2- vGlut1/MAP2
3- GABA/bIII-tubulin
4- Pre and postsynaptic marker density
We thank the reviewer for the suggestions. We don’t have immunofluorescence images quantification.
5- AD-specific deferential gene expression in PMNCs (RNAseq)
6- AD-specific deferential gene expression in Excitatory neurons (RNAseq)
We thank the reviewer for the comment. We do not have any RNASeq on PMNCs. We have now added a supplementarty Table 2 with all normalized counts in Excitatory neurons (RNAseq) for all individuals included in analysis.
7- Spontaneous neuronal activity.
We don’t have any spontaneous neuronal activity.
Round 2
Reviewer 2 Report
The Authors have addressed the majority of my concerns in their revised manuscript. The two outstanding issues are:
1) concerning Table 2 where I suggested using Allen Brain Atlas data as a source of human brain gene expression data. They do have single cell RNA-seq data here: https://celltypes.brain-map.org/rnaseq/human_m1_10x.
The second point is regarding the statement made on line 260 in the revised manuscript that "The interacting proteins were found to be enriched for those functioning in the brain". My point was that the Authors should name the brain-specific 'pathways' from their (STRING) enrichment analysis that corroborates this statement.
Author Response
In response to item 1
We thank the reviewer for pointing us to this data. Due to the nature of the data we could not just add a column of average expression levels. Instead we have added a figure with a heatmap of the same genes in human glutamatergic cells (Figure 2)
In response to item 2
We report an enrichment for genes expressed in the brain from the "tissue expression" report of STRING. There were no significant enrichments for specific pathways. We regret the misunderstanding and we have edited the text to be more clear on that point.
Reviewer 3 Report
The authors answered my question. However, if they can add immunohistochemistry, it would make more manuscripts more interesting.
Author Response
Thank you for this comment. Unfortunately at this time we are not able to perform this experiment.